# Naphthoquinone-derivative as a synthetic compound to overcome the antibiotic resistance of methicillin-resistant *S. aureus*

Ronghui Song[1], Bing Yu[2], Dirk Friedrich[1], Junfeng Li[1], Hao Shen[1], Harald Krautscheid[3], Songping D. Huang[1,4 ✉] & Min-Ho Kim [2,4 ✉]

The treatment of *Staphylococcus aureus* (*S. aureus*) infections has become more difficult due to the emergence of multidrug resistance in the bacteria. Here, we report the synthesis of a lawsone (2-hydroxy-1,4-naphthoquinone)-based compound as an antimicrobial agent against methicillin-resistant *S. aureus* (MRSA). A series of lawsone-derivative compounds were synthesized by means of tuning the lipophilicity of lawsone and screened for minimum inhibitory concentrations against MRSA to identify a candidate compound that possesses a potent antibacterial activity. The identified lawsone-derivative compound exhibited significantly improved drug resistance profiles against MRSA compared to conventional antibiotics. The therapeutic efficacy of the compound was validated using murine models of wound infection as well as non-lethal systemic infection induced by MRSA. Our study further revealed the multifaceted modes of action of the compound, mediated by three distinctive mechanisms: (1) cell membrane damage, (2) chelation of intracellular iron ions, and (3) generation of intracellular reactive oxygen species.

[1] Department of Chemistry and Biochemistry, Kent State University, Kent, OH 44240, USA. [2] Department of Biological Sciences, Kent State University, Kent, OH 44240, USA. [3] Institute of Inorganic Chemistry, Universität Leipzig, Johannisallee 29, 04103 Leipzig, Germany. [4]These authors contributed equally: Songping D. Huang and Min-Ho Kim. ✉email: shuang1@kent.edu; mkim15@kent.edu

Staphylococcus aureus (S. aureus) is a Gram-positive human pathogen causing a variety of diseases ranging from soft-tissue infections to life-threatening diseases such as endocarditis, toxic shock syndrome, and necrotizing pneumonia[1]. The treatment of these infections has become more difficult due to the emergence of multidrug resistant strains, along with their ability to evade attack by the host innate immune system, which renders the treatment of methicillin-resistant S. aureus (MRSA) critical challenges[2,3]. Most of the currently known antibiotics have been shown to confer antibacterial activities by directly interfering with the processes of translation, transcription, replication, or cell wall synthesis in bacterial cells[4]. However, bacteria have evolved multiple mechanisms to evade attacks that target those pathways via enzymatic drug modifications, increased efflux pump, or target-site resistance mutations[5,6]. Additionally, a strategy to develop antibacterial compounds by means of modifying or adding a small chemical group to conventional antibiotics had limited success in disabling the development of drug resistance, in large part due to the structural similarity between modified and existing molecules[7]. In order to reduce the probability for exposed bacteria to generate resistance phenotypes, there is an urgent need to develop antimicrobial agents that attack targets with different modes of action.

Herein, we report the synthesis of lawsone-derivative compounds as an antimicrobial agent. Lawsone is a naturally occurring hydroxynaphthoquinone (2-hydroxy-1,4-naphthoquinone), a member of the naphthoquinone family, and has been shown to exhibit biological activities including antifungal, antitumor, and antiviral activities[8–11]. Our rationale for choosing lawsone as the starting pharmacophoric platform for synthesizing the class of antimicrobial compounds is three-fold. First, lawsone is a potent and versatile Michael-acceptor that allows for the mild formation of C-C bonds at the C3 position to generate a variety of lawsone derivatives via straightforward synthetic steps in organic chemistry[12,13]. Furthermore, the substitution of the H atom at the C3 position with an R group may prevent lawsone from unselectively binding to nucleophiles such as N atoms in DNA bases and thiol groups in proteins. Second, such derivatization can preserve the essential quinonoid structural feature with the ability to undergo the stepwise one-electron reduction to semiquinone, and then to hydroquinone. This activity can catalyze the redox cycles between NADH and molecular oxygen ($O_2$) to produce intracellular reactive oxygen species (ROS)[8–11], which confers antimicrobial activity. Third, the presence of an α-hydroxy ketone moiety makes lawsone a unique chelating ligand[14,15]. Because the two O-donor atoms are hard Lewis bases, lawsone can selectively bind to the $Fe^{3+}$ ion, the hardest Lewis acid among all the other intracellular bio-essential metal ions. Since iron homeostasis is critical for the survival of bacterial cells, interception of intracellular transitory iron or demetallation of iron-containing proteins and enzymes by such chelating ligand can be detrimental for bacterial cells[16,17]. However, these unique characteristics of lawsone have not been explored for developing new antimicrobial compounds to the best of our knowledge.

In this study, a series of lawsone derivatives were synthesized by means of systematically tuning lipophilicity and screened for minimum inhibitory concentrations (MICs) against S. aureus to identify a candidate compound that exhibits a potent bactericidal activity. The therapeutic efficacy of the identified compound was validated using murine models of skin wound infection as well as non-lethal systemic infection induced by MRSA. Our study revealed that the lawsone-derivative could disable the development of resistance with multifaceted modes of action. Thus, our findings present an opportunity to utilize the unique mode of action of lawsone derivative in the development of synthetic compounds as an antibacterial strategy that can decrease the likelihood of bacterial resistance.

## Results

**Synthesis and characterization of lawsone derivatives.** The membrane of S. aureus strain is considered to be more hydrophobic and less negatively charged in general, compared to other Gram-positive strains of bacteria including Staphylococcus epidermidis[18]. With the intrinsic properties of lawsone to exhibit activities of metal chelation and ROS generation, we reasoned that fine-tuning of the lipophilicity of the lawsone molecule would exert an improved antimicrobial activity against S. aureus by facilitating the penetration of the compound through the lipid membrane of the bacteria (Fig. 1a). For this, a series of lawsone derivatives were synthesized by coupling aromatic rings comprising a lipophilic tail of varying length of carbon chains to the C3 position in lawsone via an organocatalytic three-component reductive alkylation (TCRA) reaction[19,20] (Fig. 1b). Our preliminary screening assay revealed that increasing the number of aromatic rings did not improve their antimicrobial activity against methicillin-sensitive S. aureus (MSSA). Thus, in the subsequent screening assay, a single aromatic ring comprising a lipophilic tail with varying length of carbon chains were the foci of synthetic studies. The spectroscopic and structural characterization for the entire series of lawsone derivatives performed using proton nuclear magnetic resonance ($^1$H NMR), mass spectrometry (MS) and single-crystal X-ray structural analysis confirmed that the lipophilic moiety was covalently conjugated to the C3 position by the formation of a C-C single bond (Supplementary Figs. 1–5). Among them, compound **6c** (hereafter **6c**) was found to exhibit the strongest antimicrobial activity against MRSA strains (ATCC BAA-44 and ATCC BAA-1717, MIC = 1.25–2.5 μg/mL), which was comparable to that of vancomycin and daptomycin (MIC = 1 μg/mL) (Table 1). The **6c** was also effective against vancomycin-intermediate S. aureus (VISA) (ATCC 700699) that is also non-susceptible to daptomycin (MIC = 4 μg/mL), with the same MIC value against MRSA. The determination of lipophilicity (cLogP), molecular globularity, and MIC values revealed the existence of optimal degree of lipophilicity and molecular globularity that might confer its potent efficacy against MRSA. The single-crystal X-ray structural analysis of **6c** confirmed that the lipophilic moiety with an n-buthylphenyl group was covalently conjugated to the C3 position by the formation of a C-C single bond (Fig. 1c and Supplementary Fig. 3).

**Time-kill assay and selectivity of 6c against MRSA.** Upon identifying **6c** as the most potent antimicrobial candidate from the series of derivatives, we next evaluated the antibacterial activity kinetics of **6c** using a time-kill assay on MRSA (ATCC BAA-44). The time-kill assay showed that **6c** exhibited a bacteriostatic activity at the dose of 4×MIC and bactericidal effect at 8×MIC towards MRSA (Fig. 2a). In addition to the potency of **6c** against planktonic phase of MRSA, **6c** could significantly suppress the formation of biofilm formed by MRSA at a dose of 1×MIC (5 μM or 1.5 μg/mL, Supplementary Fig. 6a–c). However, it required much higher dose of **6c** (256 μg/mL) for the eradication of established biofilm, which was comparable to those of vancomycin and daptomycin (Supplementary Fig. 6d). The selectivity of **6c** for S. aureus vs mammalian cells was assessed by determining cellular toxicity and hemolysis of **6c** against human dermal fibroblasts (HDFs) and mouse red blood cells (mRBCs), respectively. The half maximum concentration ($IC_{50}$) of **6c** against HDFs was measured to be greater than 125 μg/mL (Fig. 2b). The scanning electron microscopy (SEM) images confirmed that the treatment of **6c** (at a dose of 4×MIC for MRSA) did not alter the morphology of mRBCs (Fig. 2c) and the hemolytic concentration-50% ($HC_{50}$) was measured to be 250 μg/

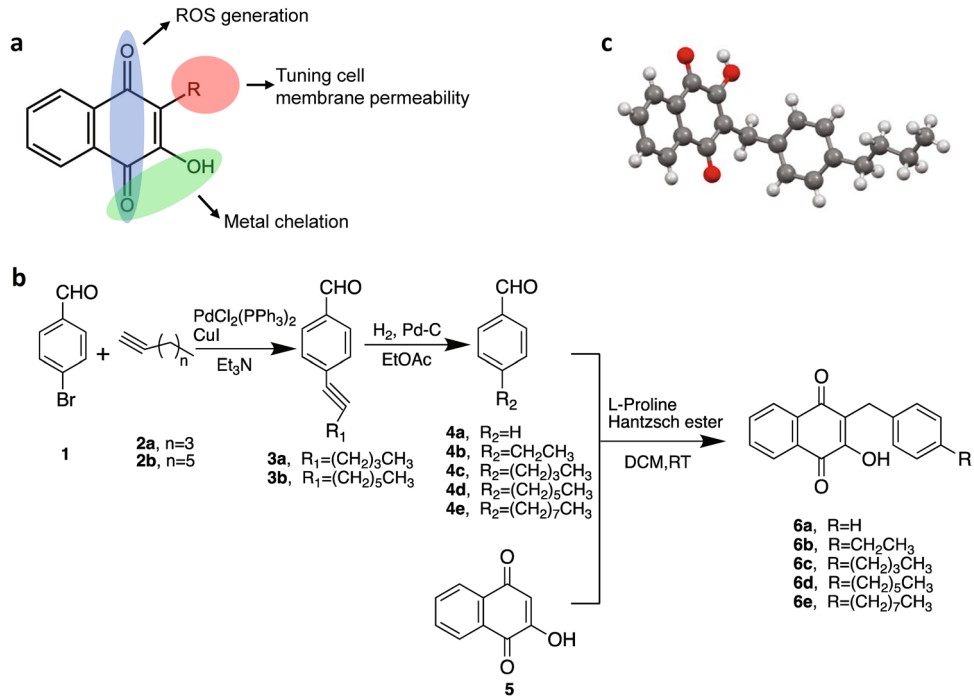

**Fig. 1 Design and synthesis of naphthoquinone derivatives. a** A schematic on the characteristics of naphthoquinone derivatives that confer antimicrobial activities. **b** Synthetic routes for the synthetic naphthoquinone derivatives including compounds **6a**, **6b**, **6c**, **6d**, and **6e**. **c** Molecular structure of compound **6c** determined from the single-crystal X-ray analysis. Red: oxygen atom; Gray: carbon atom; White: hydrogen atom.

**Table 1 MIC values, lipophilicity, and globularity of Lawsone derivatives against different strains of *Staphylococcus aureus*.**

| Compound | MIC (μg/mL) | | | | | ClogP[a] | ClogD$_{7.4}$[b] | Number of rotatable bonds[c] | Globularity[d] | Molecular weight[e] (g/mol) |
|---|---|---|---|---|---|---|---|---|---|---|
| | MSSA (ATCC 29213) | MSSA (ATCC 6538) | VISA (ATCC 700699) | MRSA (USA300, ATCC BAA-1717) | MRSA (ATCC BAA-44) | | | | | |
| Lawsone | 16 | 32 | 64 | 128 | 32 | 1.20 | −1.7 | 0 | 0 | 174.16 |
| **6a** | 128 | >128 | 128 | >128 | >128 | 3.12 | 0.6 | 2 | 0.096 | 264.28 |
| **6b** | 32 | 32 | 128 | 64 | 32 | 3.85 | 1.3 | 3 | 0.093 | 292.33 |
| **6c** | 1.25–1.9 | 0.6 | 1.25–2.5 | 1.25–2.5 | 1.25–2.5 | 4.47 | 1.9 | 5 | 0.12 | 320.39 |
| **6d** | >128 | >128 | >128 | >128 | >128 | 6.23 | 3.7 | 7 | 0.086 | 348.44 |
| **6e** | >128 | >128 | >128 | >128 | >128 | 7.45 | 4.9 | 9 | 0.060 | 376.50 |
| Vancomycin | 0.5 | 1 | 4 | 1 | 1 | −2.04 | −8.48 | 13 | 0.28 | 1449.27 |
| Daptomycin | 0.5 | 1 | 4 | 1 | 1 | −4.07 | −9.72 | 35 | 0.38 | 1620.69 |
| Ofloxacin | 0.25 | 0.25 | 16 | 0.25 | 8 | 0.83 | −1.1 | 2 | n.d. | 361.37 |
| Ciprofloxacin | 0.25 | 0.25 | 32 | 0.25 | 16 | −0.29 | −2.5 | 3 | 0.07 | 331.34 |

[a]ClogP = calculated LogP, generated using ACD/Percepta software.
[b]ClogD$_{7.4}$ = calculated LogD at pH = 7.4, generated from ACD/Percepta software.
[c]Number of rotatable bond was generated from ACD/Percepta software.
[d]Globularity was calculated from Molecular Operating Evironment (MOE).
[e]Molecular weight was calculated from ChemDraw Professional 16.0.
*n.d.* not determined.

mL (Fig. 2d). The selectivity index value of **6c** on both MSSA and MRSA was calculated to be greater than 200 against both mRBCs and HDFs, while the selectivity index value of underivatized lawsone was ~3 (Fig. 2e).

**The effect of 6c on the membrane integrity of MRSA.** We next examined whether enhanced antimicrobial effect of **6c** was associated with alteration of bacterial membrane integrity. This was assessed by quantifying the uptake of propidium iodide (PI), a membrane impermeable dye, in MRSA (ATCC BAA-44) following the treatment of **6c** (at a dose of 16 μM, 4×MIC) to the

bacteria for 60 min and the result was compared with a group of cells treated with either underivatized lawsone or vancomycin at the same concentration (16 μM). The treatment of **6c** on MRSA significantly increased the uptake of PI dye, as assessed by increase in PI fluorescence, suggesting a membrane-lytic ability of the compound, while the treatment of lawsone or vancomycin at the same concentration had little effect on the uptake of PI (Fig. 3a, b). A parallel experiment to visualize the morphological structure of the cell surfaces by SEM further revealed that the bacteria exposed to **6c** were associated with significant rupture of the cell membrane, as evidenced by the leakage of intracellular content (Fig. 3c). This result is comparable to that of daptomycin,

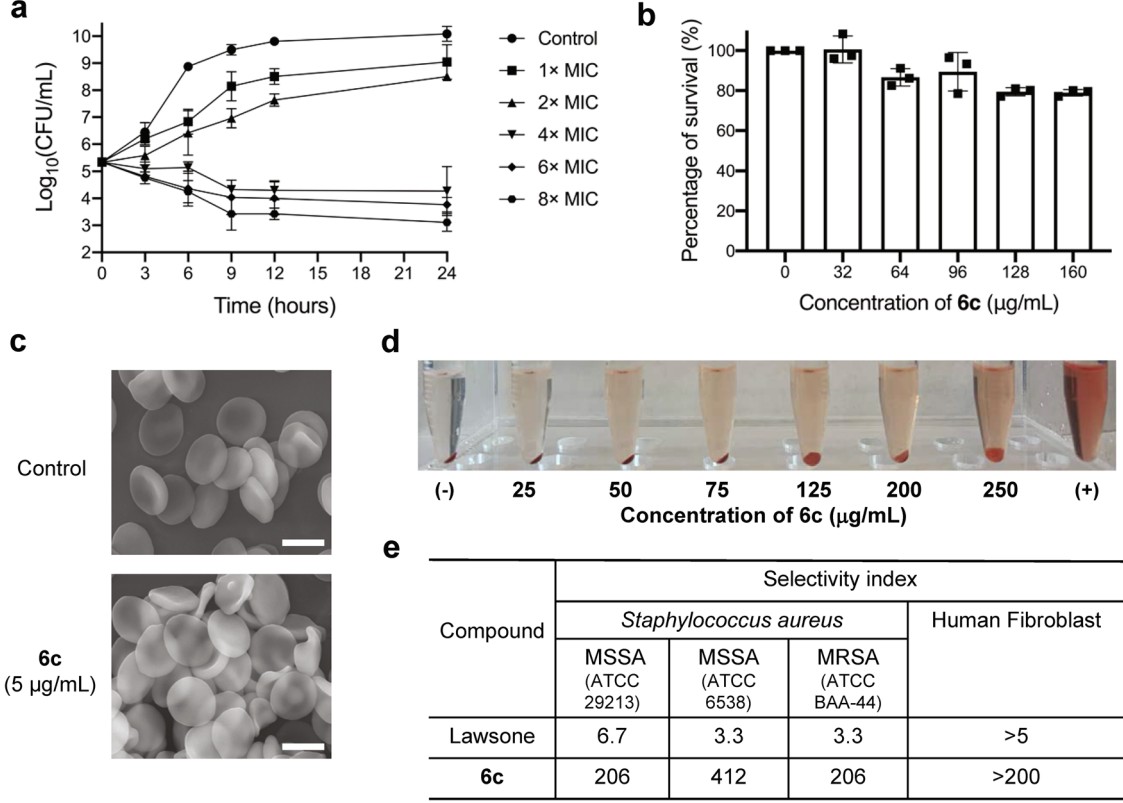

**Fig. 2 Antimicrobial efficacy and selectivity of 6c. a** Time-dependent killing of MRSA by **6c** with varying concentrations (mean ± s.d, $n = 3$ biological replicates). **b** The viability of human dermal fibroblast cells (hDFs) to **6c** with varying concentrations (mean ± s.d, $n = 3$ biological replicates). **c** SEM images of mouse red blood cells (mRBCs) treated with PBS or compound **6c** (4×MIC). Representative of three independent measurements. **d** Representative images for mRBCs hemolysis in response to **6c** with varying concentrations (0–250 μg/mL). Representative of three independent experiments. **e** The selectivity index (S.I) of **6c**. The selectivity index was calculated from $HC_{50}$ (for mRBCs) and $IC_{50}$ (for HDFs) divided by MIC of *S. aureus* strains (S.I = $HC_{50}/MIC$ or $IC_{50}/MIC$). Each biological replicate has three technical replicates.

a lipopeptide that targets bacterial cell membranes[21,22], used as a positive control. Additionally, this result is in line with the known mode of action of vancomycin, which exerts an antimicrobial activity through the inhibition of cell wall synthesis by binding to d-Ala-d-Ala termini of peptidoglycan precursors on the cell wall[23], with relatively little effect on membrane permeation[24,25]. The bacteria in the untreated control and lawsone-treated groups displayed a smooth surface without any sign of the leakage of intracellular contents. Collectively, our findings suggest that, although the detailed structural interactions between **6c** and bacterial membrane remains to be elucidated, **6c** possesses a membrane-lytic capability.

**The modes of action of 6c**. By observing the antibacterial potency of **6c** against MRSA, we next engaged in a study to explore the modes of the action of the compound. Given the critical role of iron metabolism in bacterial survival, along with the capacity of lawsone to chelate metal ions, in particular the $Fe^{3+}$ ion, we examined whether **6c** would alter the availability of intracellular iron in MRSA (ATCC BAA-44). The level of intracellular iron in MRSA was concomitantly decreased with exposure of increasing concentrations of **6c** (Fig. 4a), suggesting the iron chelation effect of the compound. We next examined whether the **6c**-induced chelation of intracellular iron was functionally linked to the bacterial killing. This was assessed by supplementing excessive concentration of free $Fe^{3+}$ ions to the **6c**-treated MRSA and quantifying the CFU number of viable bacteria. The co-treatment of excessive free iron with **6c** could rescue the viability of MRSA in a dose-dependent manner, in

which 10 times excessive iron could fully rescue the bacteria from **6c**-induced cytotoxic effect (Fig. 4b). The single-crystal X-ray structure determination revealed that **6c** might chelate intracellular iron by forming the iron complex of $[Fe(6c)_2(OH)]_4$ in the bacterial cell, which has a tetranuclear structure with 4 bridging hydroxides (Fig. 4c and Supplementary Fig. 7a). The identity of the iron-**6c** complex was further confirmed by matrix-assisted laser desorption/ionization-mass spectrometry (MALDI-MS) analysis (Supplementary Fig. 7b). Since the presence of 1,4-quinone group on the aromatic system of the molecule was shown to trigger redox signaling, we subsequently sought to examine whether **6c** could generate ROS and, if then, whether this could contribute to bacterial killing. Indeed, the treatment of **6c** significantly increased the intracellular level of ROS in MRSA (Fig. 4d). Scavenging of ROS by superoxide dismutase (SOD) could rescue the viability of MRSA partially (Fig. 4e). Collectively, our results demonstrate that our compound **6c**, along with its ability to damage bacterial cell membrane, could exhibit an antimicrobial activity through the chelation of intracellular iron as well as generation of ROS once it enters the cytoplasm.

Since the molecular targets of **6c** are not overlapped with those of conventional antibiotics, we next examined whether the compound could synergize the activity of oxacillin, a methicillin class of antibiotics, against MRSA. The MIC of oxacillin was measured to be 64 μg/mL and the combined treatment of oxacillin with 1/4 MIC of **6c** could significantly reduce the MIC of oxacillin from 64 μg/mL to below 4 μg/mL against MRSA (Supplementary Fig. 8), which resulted in the fractional inhibitory concentration (FIC) index of 0.31, suggesting the synergistic effect of **6c** and oxacillin.

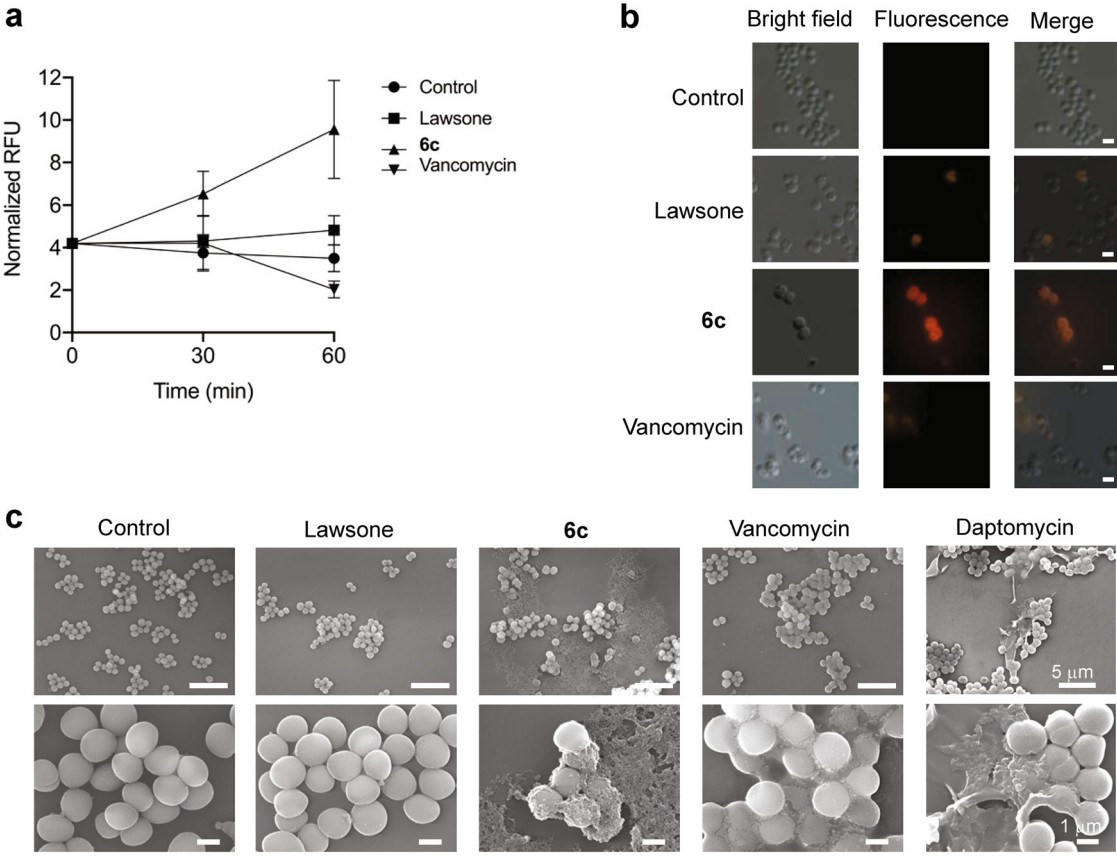

**Fig. 3 Effects of compound 6c on the membrane permeability and damage in MRSA. a** The change in PI fluorescence in MRSA treated with lawsone, vancomycin, or **6c** (all treated at the 16 μM concentration) over the duration of 1 h (mean ± s.d, $n = 3$ biological replicates and each biological replicate has three technical replicates). **b** Representative microscopic images of PI in MRSA treated with lawsone, vancomycin, or **6c** (all at 16 μM) for 1 h (Scale bar: 2 μm). Differential interference contrast microscopic (DIC) images (left), fluorescence images (middle), and merged images of MRSA (right). Representative of three independent experiments. **c** Representative SEM images of MRSA treated with lawsone, **6c**, vancomycin, or daptomycin (all at 16 μM) for 2 h. Representative of three independent experiments.

**In vitro evaluation for resistance development**. We next examined the potential for the development of resistance through serial exposure of *S. aureus* to **6c** for successive passages. The MIC values for wild-type *S. aureus* strain were determined first. Then, cells grown in the initial MIC dose of the compound or antibiotics from the previous passage were once again harvested after 24-h incubation period and assayed for a new MIC. The process was repeated for 56 passages against MRSA. The initial MIC values of ofloxacin, vancomycin and **6c** against MRSA were 8.0, 1.0 and 1.25 μg/mL, respectively. The MIC of ofloxacin increased by 4-fold after 5 passages, by 32-fold at 35 passages, and 64-fold at 56 passages (Fig. 5a). The MIC of vancomycin was modestly increased by 4-fold by 51 passages and reached an abrupt increase of 128-fold at 56 passages by displaying a mutant phenotype of MRSA resistant to vancomycin (MRSA[vanR]). In contrast, the MIC of **6c** was largely unchanged up to 33 passages and then modestly increased by only 4-fold, which was maintained by 56 passages, the end of measurements. The improved drug resistance profile of **6c** was also observed for MSSA (ATCC 29213 strain) (Supplementary Fig. 9). We next examined whether **6c** could still retain antimicrobial activity against mutant strains of MRSA developed with repeated exposures of ofloxacin and vancomycin. The MICs of **6c** against mutant MRSA that had become resistant to ofloxacin (MRSA[oflR]) or vancomycin (MRSA[vanR]) at the day 56 passage were not substantially altered from initial value of MIC for wild-type MRSA, which measured to be unchanged for MRSA[oflR] (1.25 μg/mL) and 5 μg/mL for

MRSA[vanR] (Fig. 5b and Supplementary Fig. 10). This suggests that our compound **6c** did not generate cross-resistance against mutant *S. aureus*. Collectively, it appears that the capacity of **6c** to exhibit multifaceted modes of action might contribute to disable the development of resistance against MRSA (Fig. 5c).

**In vivo validation using murine models of infection**. By observing the capacity of compound **6c** in promoting the bactericidal activity against MRSA in vitro, we immediately sought to validate its efficacy in vivo using murine models of local wound infection as well as non-lethal systemic infection induced by MRSA. For local infection, C57BL/6 mice were inoculated with $1 \times 10^6$ CFU MRSA in the wounded skin at day 0 and then a defined amount of **6c** or vancomycin (4 mg/kg in 40 μL sterile saline per wound) was topically applied once to the wounds of mice at day 1 post-infection. The CFU number of viable bacteria in the wound was quantified at day 2 post-infection (Supplementary Fig. 11a). As compared with the control group, the single dose treatment of **6c** significantly reduced *S. aureus* numbers in the wound by 60%, which was comparable to that of vancomycin (~50% reduction in bacterial burden in wounds) (Fig. 6a). For validation of the compound for systemic infection, MRSA ($1 \times 10^8$ CFU in 100 μL PBS) were intraperitoneally injected (i.p) to C57BL/6 mice and the mice were treated with a single dose of **6c** (15 mg/kg, i.p) at day 1 post-infection. The CFU number of viable bacteria was counted at day 2 post-infection from major organs including lung, kidney, spleen, liver, and peritoneal tissues

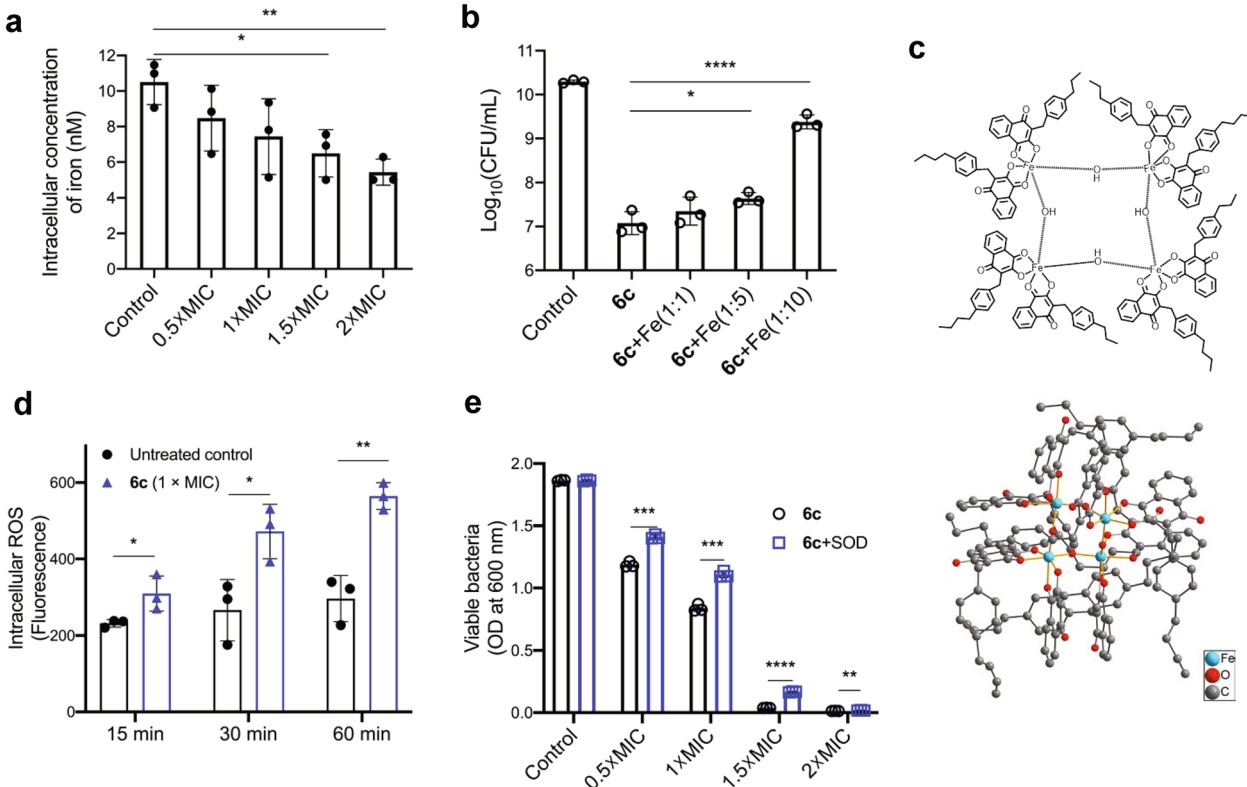

**Fig. 4 Effects of compound 6c on the iron chelation and ROS generation in MRSA. a** Intracellular concentration of iron from MRSA treated with varying concentrations of **6c** (mean ± s.d, $n = 3$ biological replicates). **b** The effect of supplementing iron ions (FeCl$_3$) on rescuing **6c**-mediated killing of MRSA for varying ratios of **6c** (7.5 μg/mL) and Fe$^{3+}$ (1:0, 1:5, and 1:10) (mean ± s.d, $n = 3$ biological replicates). **c** Top: chemical structure of iron-**6c** complex, [Fe(**6c**)$_2$(OH)]$_4$; Bottom: molecular structure of iron-**6c** complex from single crystal X-ray diffraction. **d** Intracellular level of ROS from MRSA treated with 1×MIC concentration of **6c** (mean ± s.d, $n = 3$ biological replicates). **e** The effect of superoxide dismutase (SOD) on rescuing **6c**-mediated killing of MRSA treated with varying concentrations of **6c** (mean ± s.d, $n = 3$ biological replicates). $*p < 0.05$, $**p < 0.01$, $****p < 0.001$, and $****p < 0.0001$.

(Supplementary Fig. 11b). The mice treated with **6c** exhibited significantly reduced bacterial burdens in kidney, liver, and peritoneal tissues, compared to vehicle control group ($p < 0.05$) (Fig. 6b). Collectively, our results validate that compound **6c** is therapeutically effective in eradicating MRSA in both local and systemic infections.

## Discussion

In this study, we report the synthesis of a lawsone-based synthetic antimicrobial compound (**6c**) that exhibits a potent antimicrobial activity as well as improved drug resistance profile against MRSA. The therapeutic efficacy of the compound was validated using murine models of local as well as systemic infections induced by MRSA. The findings of our study further revealed the multi-faceted modes of action of compound **6c**, mediated by three distinctive mechanisms: (1) cell membrane damage, (2) chelation of intracellular iron ions, and (3) generation of intracellular ROS.

Our findings support that lawsone molecule can serve as a building-block material for the development of new synthetic antimicrobial agents. Although the biological activities of lawsone as an anti-tumor or anti-fungal agent have been reported, its use as an antimicrobial agent has been limited due to its relatively high value of MIC against *S. aureus* (MIC of ~32 μg/mL) and lack of selectivity. Based on the hydrophobic nature of *S. aureus* membrane amongst other Gram-positive strains, we hypothesized that fine-tuning of the lipophilicity of lawsone would significantly improve the antimicrobial activity of the compound against *S. aureus* by facilitating its penetration through the cell membrane. An underivatized form of lawsone is

associated with a low hydrophobicity (clogP of 1.20) and a relatively planar structure (globularity of 0), which might limit its favorable interactions with bacterial lipid membrane. Our results reveal the existence of optimal degree of lipophilicity and molecular globularity, which are neither too high nor too low, for achieving the enhancement of bacterial membrane permeability. The results of **6c** on rotatable bonds of 5 and globularity of 0.12 are similar to those of conventional antibiotics such as penicillin G (rotatable bonds = 4, globularity = 0.17), ampicillin (rotatable bonds = 4, globularity = 0.12), moxifloxacin (rotatable bonds = 4, globularity = 0.15), and ranbezolid (rotatable bonds = 7, globularity = 0.12)[26]. Although the structural moiety that **6c** binds in the membrane surface of *S. aureus* yet to be elucidated, the compound appears to penetrate across the lipid layer by causing irreversible damage to the membrane, as evidenced by the significant leakage of intracellular content. Since underivatized lawsone could not disturb bacterial membrane, the degree of lipophilicity and globularity maintained by the substituent R group including the aliphatic chain of four carbon atoms in **6c** seems to provide a balanced moiety for the favorable intermolecular interactions with the lipid membrane of *S. aureus*, leading to subsequent damage and rupture of the membrane. Taken together, these results support that the lipophilic moiety in **6c**, associated with certain degree of lipophilicity and molecular globularity, might facilitate its binding to lipid membrane of MRSA and subsequently cause irreversible damage to the membrane.

It has been largely considered that natural antibiotics exhibit superior selectivity in targeting bacterial cells over synthetic compounds. However, bacteria were shown to rapidly develop

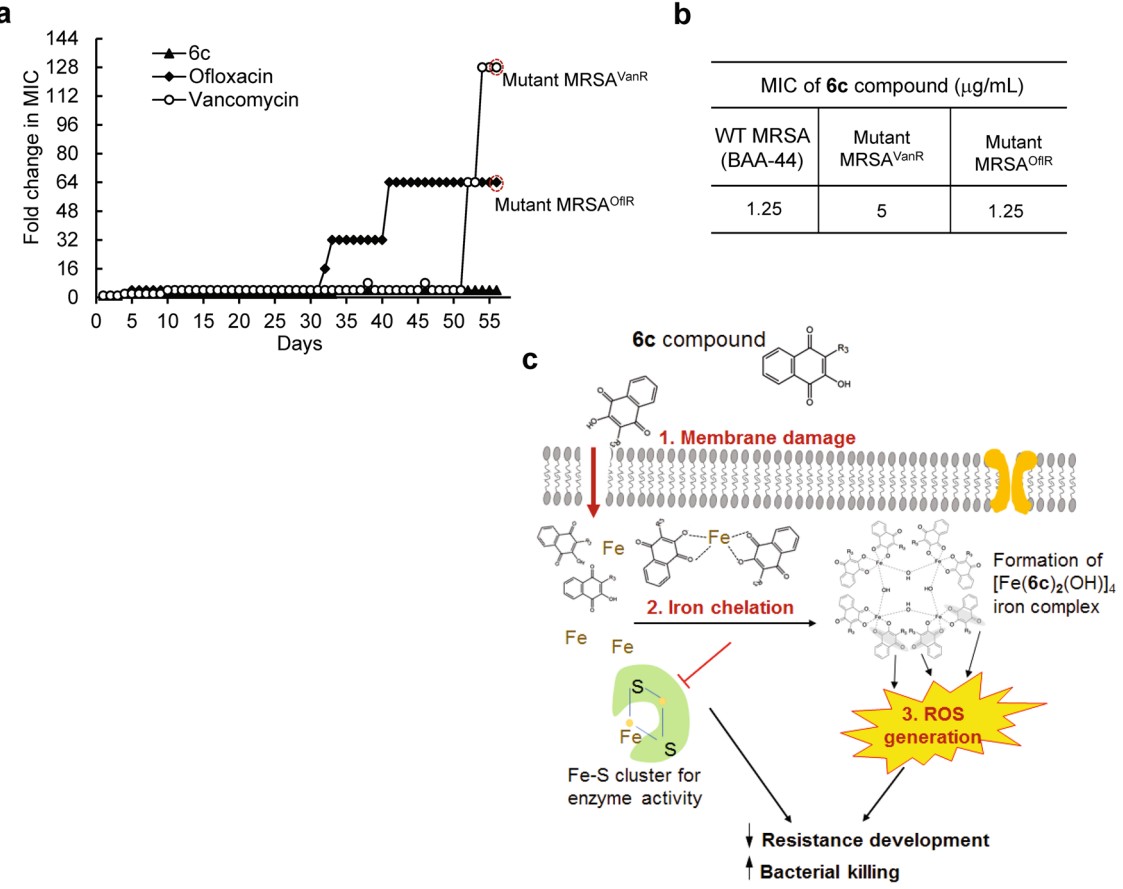

**Fig. 5 In vitro assay for the development of resistance in MRSA towards 6c. a** The development of resistance in serial passage of MRSA with repeated exposure of ofloxacin, vancomycin, and **6c** over 56 days. The initial MIC values of 8 μg/mL, 1 μg/mL, and 2.5 μg/mL, were used for ofloxacin, vancomycin, and **6c**, respectively. This assay was repeated three times independently yielding similar results. **b** MIC values of **6c** against MRSA that become resistant to ofloxacin (mutant MRSA^oflR) and vancomycin (mutant MRSA^vanR). **c** A schematic on the modes of antimicrobial actions of **6c** against MRSA.

their resistance mechanisms to evade natural antibiotics or even their chemically modified forms[27]. This may be associated with that most of natural antibiotics exhibit antibacterial activities by interfering with the basic biological process of replication, transcription, and translation, in which bacteria could rapidly develop into resistant mutants via natural selection. Our results demonstrate a significantly improved drug resistance profile of compound **6c** against MRSA over conventional antibiotics including ofloxacin and vancomycin. This appears to be the capacity of compound **6c** to exhibit an antimicrobial activity with multifaceted modes of action. In particular, the antibacterial mechanism of action of compound **6c** by means of disrupting the homeostasis of metal ions is fundamentally different from that of conventional antibiotics. As a cofactor of many vital enzymes including those involved in cellular respiration and DNA synthesis, iron is a vital nutrient for nearly all forms of life, including bacteria[28,29]. Although, iron homeostasis has been explored as an antimicrobial target, prior efforts have primarily focused on the inhibition or interference with iron-acquisition proteins, such as siderophores[30,31]. Many pathogens, however, develop mechanisms to diminish the efficacy of the iron acquisition pathway[32]. In contrast, a strategy for direct chelation of metal ions, as evidenced by certain antibiotics such as dithiolopyrrolones, could disrupt the cellular processes of bacteria by altering metal availability as well as interfering with activities of metalloenzymes[33]. Additionally, we recently reported that a strategy to chelate intracellular free iron with nanoparticles capable of iron exchange could be an effective antimicrobial strategy against *S. aureus*[34].

Although bacteria have evolved various means of evading the mode of actions by traditional antibiotics, so far, they have failed to develop resistance for the mechanism of disrupting metal ion homeostasis. As an antibacterial strategy that can decrease the likelihood of bacterial resistance, this unique mode of action can be further explored for the treatment of multidrug-resistant infections. In particular, since the mode of antimicrobial action of our lawsone-based compound does not overlap with those of traditional antibiotics, the lawsone-based drugs can complement traditional antibiotics and may be used as a combination therapy against multidrug resistant bacterial infections as supported in our checkerboard assay (Supplementary Fig. 8).

In summary, we have synthesized a synthetic antimicrobial compound that exhibits a potent antimicrobial activity and improved resistance profile with multiple modes of action against MRSA. Our findings strongly support that compound **6c** is a promising lead candidate for further development as an antimicrobial agent for the treatment of *S. aureus* infection associated with resistance development.

## Methods

**Synthesis and characterization of lawsone-derivatives**. The chemical reagents used in the synthesis of lawsone-derivatives were purchased from Sigma-Aldrich, Acros Organic, Alfa Aesar, TCI America, and Fisher Scientific without further purification. Deuterium solvents were purchased from Cambridge Isotope. Detailed information on the synthesis and characterization of lawsone-derivatives are given in supplementary methods.

Bacteria strains, growth media and antibiotics. The bacterial strains of MSSA (ATCC 29213 and ATCC 6538), MRSA (ATCC BAA-44 and ATCC BAA-1717),

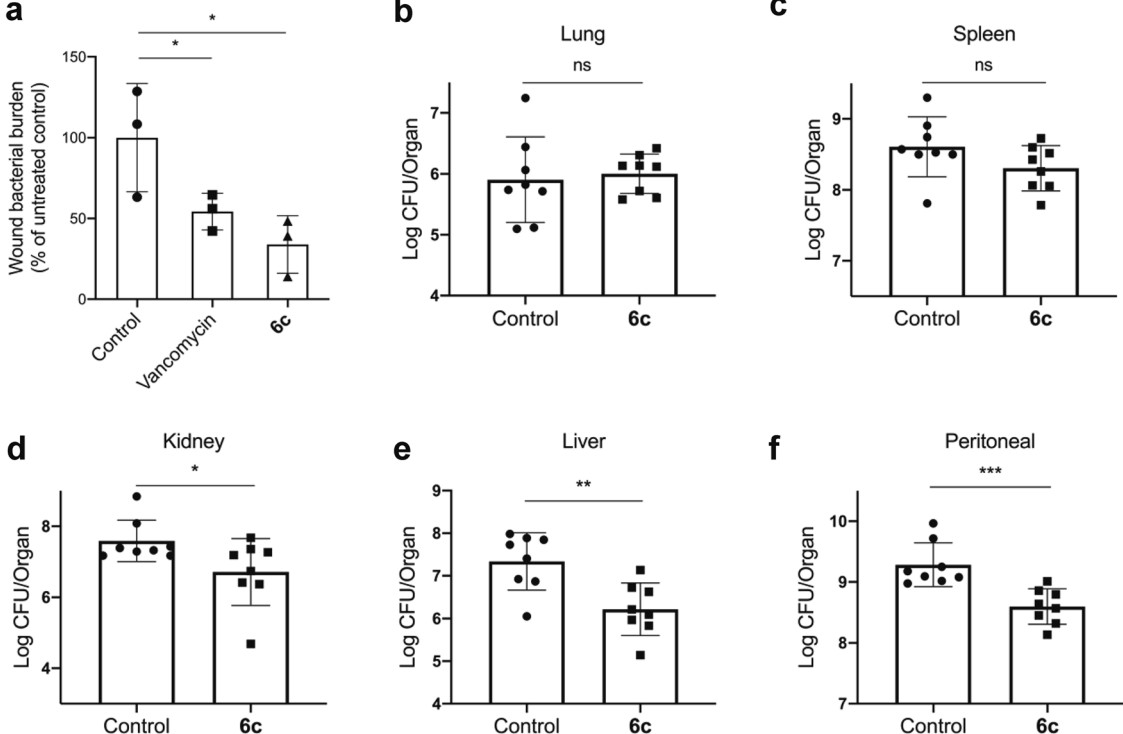

**Fig. 6 In vivo antibacterial efficacy of 6c against MRSA in murine models of skin wound infection and peritoneal infection. a** The effect of the treatment of **6c** or vancomycin (via topical treatment, single dose at 4 mg/kg) on the bacterial burden in wounds of mice infected by MRSA at day 2 post-wounding (mean ± s.d, $n = 3$ mice per group) *$p < 0.05$ vs control. **b–f** The effect of the treatment of **6c** or vancomycin (via i.p injection, single dose at 15 mg/kg at day 2 post-infection) on the bacterial burden in lung (**b**), spleen (**c**), kidney (**d**), liver (**e**), and peritoneal (**f**) tissues of mice infected by MRSA (mean ± s.d, $n = 8$ mice per group). The CFU counts from homogenized tissue collected from each mouse was run in duplicates. *$p < 0.05$, **$p < 0.01$, and ***$p < 0.001$.

and VISA (ATCC 700699) were purchased from American Type Culture Collection. The mueller hinton broth (MHB) and tryptic soy broth (TSB) powder were purchased from Fisher Scientific. Antibiotics including ciprofloxacin, ofloxacin, oxacillin and vancomycin were purchased from Sigma Aldrich. Daptomycin was purchased from TCI America.

**Mouse models of wound infection and peritoneal infection by MRSA**. C57BL/6 mice (male mice, 8–12 weeks old) were obtained from Jackson Laboratory and used for both infection models of local wound infection and non-lethal peritoneal infection. The experimental protocol was reviewed and approved by the Institutional Animal Care and Use Committee of Kent State University. For local wound infection in mouse, a full-thickness circular wound was made on the dorsal skin of mouse using a 6 mm sterile biopsy punch (Acuderm Inc.)[35,36]. The wound was covered with a transparent, semipermeable Tegaderm dressing (3 M) and $1 \times 10^6$ CFU of MRSA (ATCC BAA-44) was inoculated under the dressing to the wounded skin and randomly divided into three groups. At day 1 post infection, 40 μL of sterile saline, vancomycin (4 mg/kg in 40 μL of saline), or **6c** (4 mg/kg in 40 μL of saline) was locally injected to the area of the wound infection. At day 2 post-infection, mice were euthanized with $CO_2$ inhalation and wounded skin was excised and homogenized for bacterial CFU counting.

For the induction of non-lethal peritoneal infection, a solution of MRSA ($1 \times 10^9$ CFU/mL in 100 μL, ATCC BAA-44) was intraperitoneally injected into the mice at day 0 and randomly divided into two groups. At day 1 post-infection, mice were treated with single dose of either sterile saline (100 μL) or **6c** (15 mg/kg in 100 μL) via intraperitoneal injection. At day 2 post-infection, mice were euthanized with $CO_2$ inhalation and organs, including lung, liver, kidney, spleen, peritoneal tissue, were excised and homogenized for bacterial CFU counting.

**Determination of minimum inhibitory concentration (MIC)**. MIC was determined by the broth microdilution method by following Clinical and Laboratory Standards Institute (CLSI) guidelines[37]. The tested bacterial strains were cultured in MHB (for ATCC 29213 strain of *S. aureus*) or TSB (for ATCC 6538 strain, ATCC BAA-44, ATCC BAA-1717 and ATCC 700699 strains of *S. aureus*). A varying concentration of lawsone-derivatives or conventional antibiotics were dissolved in the media with *S. aureus* ($5 \times 10^5$ CFU/mL). The resulting suspensions were transferred to a 96-well microliter plate at 200 μL/well (three wells for each compound). The plate was then incubated at 37 °C for 20 h. MIC values were then determined as the lowest concentration of compound at which no visible growth of bacteria was observed.

**SEM imaging of MRSA and mouse red blood cells**. The morphology of MRSA was characterized by SEM[38]. Briefly, MRSA ($1 \times 10^9$ CFU/mL, ATCC BAA-44) were treated with lawsone, **6c**, vancomycin, or daptomycin (all at a 16 μM concentration) for 2 h. The bacterial solution was then centrifuged at 1320× $g$ for 6 min at 4 °C and resuspended into 1 mL of PBS twice. Subsequently, the bacteria were fixed with PBS containing 2.5% glutaraldehyde. After washing with PBS three times, the bacteria were subjected to 1 h post-fixation with 1% $OsO_4$ in PBS. For the characterization of RBC morphology using SEM, mRBCs were isolated from C57BL/6 mouse as described previously. Briefly, the cells were diluted with PBS to reach a final concentration of approximately $10^6$ erythrocytes/mL and treated with lawsone, **6c**, or vancomycin at the same concentration of 16 μM at 37 °C for 1 h. After incubation, the solution was resuspended into a 1 mL of PBS twice and then fixed with PBS containing 2.5% glutaraldehyde. After washing with PBS three times, the RBC were subjected to 1 h post-fixation with 1% $OsO_4$ in PBS.

After fixation, either bacteria or mRBCs samples were washed three times with PBS and dehydrated with a series of graded ethanol solutions. The samples were then dried in the air and coated by gold. SEM images were taken by Quanta450 Field Emission Gun Scanning Electron Microscope (FEG SEM).

**Cell membrane permeabilization assay**. The bacterial membrane permeabilization assay was performed using fluorescence microscopy and spectrophotometric analysis[39]. Briefly, MRSA ($1 \times 10^9$ CFU/mL, ATCC BAA-44) were treated with lawsone, **6c**, and vancomycin (all at a 16 μM concentration) for 2 h at 37 °C with agitation (180 rpm). After incubation, cells were centrifuged and resuspended with PBS as described above. Propidium iodide (PI) was then added at a final concentration of 20 μM and the solution was incubated for 10 min at 37 °C in the dark. Glass slides immobilized with 8 μL of each mixture were observed under an inverted microscope (Olympus IX83). Both the differential interference contrast images and fluorescence images were taken with a 100× objective (Olympus UAPO N 100x/1.49 TIRF) and captured by a digital camera (Hayear Electronics). The fluorescent probes were excited by a 532-nm laser (Coherent, OBIS LS 50 mW) and the emission signals were filtered by a set of bandpass filters (Chroma, TRF49907-ET).

For a spectrophotometric analysis, PI was added to the cells at a final concentration of 10 μM and the mixture solution was incubated for 15 min at 37 °C in the dark. After incubation, the solutions (100 μL) were immediately transferred to a 96 well plate, and the relative fluorescence unit (RFU) was measured at 535/617 (Excitation/Emission) by using a microplate reader (SpectraMax M4). The

number of viable bacteria in the sample was obtained using an agar plate counting method. The RFU value was normalized by number of viable bacteria (CFU/mL).

**Time-kill assay**. Time-kill assay was performed against MRSA[40]. The MRSA ($1\times10^5$ CFU/mL, ATCC BAA-44) grown at early exponential phase was treated with **6c** at varying concentrations ($1\times$MIC, $2\times$MIC, $4\times$MIC, $6\times$MIC and $8\times$MIC) at 37 °C. The number of viable bacteria (CFU/mL) was counted at selected time points (3 h, 6 h, 9 h, 12 h, and 24 h) by serially diluting bacterial suspension and plating it in triplicate on tryptic soy agar plate.

**Iron rescue assay**. The suspension of MRSA ($1\times10^6$ CFU/mL, ATCC BAA-44) was treated with compound **6c** (7.5 μg/mL) or mixture of **6c** and $FeCl_3$ prepared at varying ratios of **6c** to $Fe^{3+}$ ranging from 1:1, 1:5, to 1:10 for 24 h. The CFU number of viable bacteria in the sample was obtained using agar plate counting method as described above.

**Measurement of intracellular iron in MRSA**. The measurement of intracellular iron in bacterial cells was performed using an iron assay kit (ab83366, Abcam). Briefly, MRSA ($2\times10^8$ CFU/mL, ATCC BAA-44) solutions prepared in TSB medium were treated with a varying concentrations of compound **6c** ($0.5\times$ MIC, $1\times$ MIC, $1.5\times$ MIC, and $2\times$ MIC) for 1 h at 37 °C with agitation (180 rpm). The bacterial solution was then centrifuged at $1320\times g$ for 6 min at 4 °C and resuspended with PBS twice. Subsequently, the solution was transferred into a 96-well plate and incubated with iron probe for 1 h at 37 °C in the dark. The absorbance was measured at 593 nm by using a microplate reader (SpectraMax M4). The number of viable bacteria in the sample was obtained using agar plate counting method. The absorbance value was normalized by number of viable bacteria (CFU/mL).

**Measurements of intracellular ROS in MRSA**. The intracellular level of ROS in bacterial cells was determined using an Image-iT™ LIVE Green ROS Detection Kit. Briefly, MRSA ($2\times10^8$ CFU/mL, ATCC BAA-44) solution prepared in TSB medium were treated with a varying concentration of compound **6c** ($0.5\times$ MIC and $1\times$ MIC) for 1 h at 37 °C with agitation (180 rpm). The bacterial solution was then centrifuged at $1320\times g$ for 6 min at 4 °C and resuspended with HBSS. The carboxy-$H_2$DCFDA was then added to the solution at a final concentration of 20 μM and incubated for 30 min at 37 °C in the dark. After incubation, the solutions (100 μL) were immediately transferred to a 96-well plate, and RFU was measured at 495/529 (Excitation/Emission) by using a microplate reader (SpectraMax M4). The number of viable bacteria in the sample was obtained using agar plate counting method. The RFU value was normalized by number of viable bacteria (CFU/mL).

**Superoxide scavenging assay**. The MRSA ($1\times10^6$ CFU/mL, ATCC BAA-44) prepared at the early exponential phase was treated with a varying concentration of compound **6c** ($0.5\times$ MIC, $1\times$ MIC, $1.5\times$ MIC, and $2\times$ MIC) in the presence and absence of superoxide dismutase (SOD, final concentration 250 U/mL) in culture tubes at 37 °C at 180 rpm for 24 h. Then, bacterial number was assessed by measuring the absorbance of the samples at 600 nm.

**Mammalian cell viability assay**. The cytotoxicity of compound **6c** toward mammalian cells was assessed using an MTT viability assay. Human dermal fibroblast cells (ATCC PCS201010) were seeded in a 96-well plate at a density of $4\times10^5$ cells per well with DMEM high-glucose medium and incubated for 24 h at 37 °C. The cells in each well were then treated with varying concentrations of **6c** (0, 25, 50, 75, 100, and 125 μg/mL) and then incubated for 24 h. The cells were then incubated with a 10 μL of MTT reagent for 2 h at 37 °C. The viability of the cells was quantified by measuring absorbance at 593 nm by using a microplate reader (SpectraMax M4) and the results were expressed as the percentage of viable cells with respect to the viability of untreated control cells.

**Checkerboard assay**. The checkerboard assay was performed in a 96-well plate[41]. The antibiotic oxacillin was 2-fold serially diluted along the row-axis and the **6c** compound was 2-fold serially diluted along the column-axis to create a matrix in which each well consists of a combination of both at different concentrations. Each well was inoculated with MRSA (ATCC BAA-44) to yield approximately $5\times10^5$ CFU/mL in a 100-μL final volume, incubated for 20 h at 37 °C and examined for visibility to determine the MIC. Dividing the MIC of **6c** in the presence of antibiotic by the MIC of **6c** alone was used to calculate the fractional inhibitory concentration (FIC) of the **6c**. Similarly, dividing the MIC of antibiotic in the presence of **6c** by the MIC of antibiotic alone was used to calculate the fractional inhibitory concentration (FIC) of the antibiotic. The FIC index was calculated by the summation of both FIC values. The FIC index was interpreted as synergistic, additive, or antagonistic for values of $x \leq 0.5$, $0.5 < x < 4$, or $\geq 4$, respectively[42].

**Biofilm inhibition assay**. The biofilm inhibition assay was performed in a 96-well plate[43]. An overnight culture of MRSA (ATCC BAA-44) was diluted to a final cell concentration of $1\times10^6$ CFU/mL and transferred (100 μL) to a 96-well plate

containing lawsone-derivative compounds (**6a**, **6b**, **6c**, **6d**, and **6e**) at varying concentrations ranging from 0 to 30 μM. The bacterial cells were incubated with stationary phase for 24 h at 37 °C to form biofilms. After incubation, the biofilm was gently washed to keep the integrity of the biofilms intact and resuspended with PBS. For the quantification of biofilm mass, the solutions in the plate with biofilm were replaced with a 100 μL solution containing 0.5% of crystal violet and incubated for 30 min at room temperature in the dark. After incubation, the biofilms were added with PBS and solubilized by adding 150 μL acetic acid. The absorbance of the biofilm was measured at 595 nm using a microplate reader (SpectraMax M4) and the results were expressed as percentage changes with respect to the control (without compounds). For the quantification of the number of viable bacteria in the biofilm, the biofilms were gently destroyed and plated on TSA after serial dilutions of each suspension. The number of viable bacteria in the sample was obtained using agar plate counting method as described above and results were expressed as percentage changes with respect to the control (without compounds).

**Biofilm destruction assay**. The biofilm destruction assay was performed in a 96-well plate[43]. An overnight culture of MRSA (ATCC BAA-44) was diluted to a final cell concentration of $2\times10^6$ CFU/mL in TSB with 0.5% glucose and transferred (100 μL) to a 96-well plate. The plate was sealed and incubated with stationary phase for 24 h at 37 °C to form biofilms. After incubation, the plate was gently washed to keep the integrity of biofilm. Then, the stock solutions (TSB) containing vancomycin, daptomycin and **6c** at varying concentrations ranging from 4 to 512 μg/mL. The bacterial cells were incubated at stationary phase for 24 h at 37 °C. After incubation, the biofilm was gently washed three times using PBS to keep the integrity of the biofilms intact and resuspended with PBS. For the quantification of biofilm mass, the solutions in the plate with biofilm were replaced with a 110 μL solution containing 0.5% of crystal violet and incubated for 30 min at room temperature in the dark. After incubation, the biofilms were washed with PBS until fully remove crystal violet solutions and then, solubilized by adding a 150 μL acetic acid. The absorbance of the biofilm was measured at 595 nm using a microplate reader (SpectraMax M4) and the results were expressed as percentage changes with respect to the control (without compounds).

**Biofilm staining with SYTO9 and PI**. The biofilm staining assay was performed in an 8-well plate[44]. An overnight culture of MRSA (ATCC BAA-44) was diluted to a final cell concentration of $1\times10^6$ CFU/mL and transferred (300 μL) to an 8-well plate containing **6c** compound at varying concentrations ranging from 10 to 30 μM. compounds. The bacterial cells were incubated stationary phase for 24 h at 37 °C to form biofilms. After incubation, the biofilm was gently washed to keep the integrity of the biofilms intact and resuspended with PBS. The solutions in the plate with biofilm were then replaced with a 200 μL solution containing 0.3% of both SYTO 9 and PI (LIVE/DEAD *Bac*Light Bacterial Viability Kit, Thermo Fisher Scientific). The plate was then incubated for 20–30 min at room temperature in the dark. After incubation, every sample well was washed with PBS and the fluorescence images were acquired by using an inverted fluorescence microscopy (Olympus IX81).

**Hemolysis assay**. The hemolytic activity of **6c** was measured using RBCs. The RBCs isolated from C57BL/6 mouse were centrifuged at $1000\times g$ for 6 min at 4 °C and resuspended in PBS at a final concentration of approximately $10^7$ erythrocytes/mL. The stock solution of **6c** was serially diluted and added to the suspension of mRBCs to reach final concentrations of **6c** ranging from 25 to 250 μg/mL and $1\times 10^6$ erythrocytes/mL of RBC. The solutions with PBS and Triton X-100® (1%v/v) were used as negative and positive controls, respectively. Subsequently, the suspensions were incubated at 37 °C for 60 min. The absorbance was measured at 540 nm. The percentage of hemolysis was calculated using the following equation: % hemolysis = [absorbance of the sample – absorbance of blank] ÷ [absorbance of positive control] × 100. The concentrations of tested compounds required to lyse 50% of the erythrocytes were used to calculate the fifty percent hemolysis ($HC_{50}$) values.

**In vitro assay for resistance development**. The resistance development by sequential passaging was performed[40]. The potential for the development of resistance by compound **6c**, vancomycin, and ofloxacin through serial exposure of the compound to the MRSA (ATCC BAA-44) for serial passages, and the changes in MIC values were monitored over a period of 56 days. Briefly, initial MIC values of **6c** and antibiotics were first determined. After incubation, serial passaging was initiated by treating bacterial cells at the MIC of the compound or antibiotics and inoculating them into the fresh TSB. This inoculum was subjected to another MIC assay with agitation (180 rpm). After a 24 h incubation at 37 °C, cells grown at the highest concentration of the compound or antibiotics were once again harvested and assayed for the MIC. This process was repeated for 56 passages for MRSA. Results were expressed as a fold increase in MIC with respect to the initial value of MIC.

**Lipophilicity and globularity determination**. Number of rotatable bonds, cLogP and $cLogD_{7.4}$ values were calculated using the ACD/Percepta software (Ver 2019.1.3, Advanced Chemistry Development, Inc). Globularity was calculated using

Molecular Operating Environment 2019 (MOE 2019). The crystal data of derivatives were input to MOE (Ver 2019) and the results were generated from descriptor. The data of antibiotics (vancomycin, daptomycin and ciprofloxacin) were adapted from a reported study[26].

**Statistics and reproducibility**. Statistical analysis was performed using GraphPad Prism version 8.0 software. A two-tailed unpaired t-test was used to determine statistical significance between two groups. A statistical significance among multiple groups was analyzed using One-way ANOVA followed by Holm-Sidak comparisons test. For all analyses, p-value of less than 0.05 was considered to be statistically significant. Data were presented as mean ± standard deviation (mean ± s.d). The studies for in vitro cell culture were run in triplicate (3 biological replicates and 3 technical replicates per each biological replicate). For in vivo mice studies, mice were randomly divided into two (for non-lethal systemic infection protocol) or three (for wound infection protocol) groups and CFU counts from the collected tissue of the mice were run in duplicates.

**Reporting summary**. Further information on research design is available in the Nature Research Reporting Summary linked to this article.

## Data availability
The XRD crystal data have been deposited in the Cambridge Structural Database (Deposition Number: 1989180, 1989181, 1989182, 1989183, 1989184). All source data underlying the graphs presented in the main figures are available in the Supplementary Data 2. Other data or information that support the findings of this study are available from the corresponding author upon request.

## Code availability
No custom code or mathematical algorithms were used in this study.

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

## Acknowledgements
This work was funded by the National Institutes of Health (R01NR015674). We thank G. Chen, L. Li and Z. Wang for the technical assistance for cellular studies, K. Wang for helpful discussion regarding chemical synthesis, M. Gao for help with SEM analysis. The SEM used in this study was conducted at Advanced Materials and Liquid Crystal Institute at Kent State University.

## Author contributions
R.S. designed and synthesized the naphthoquinone derivatives and contributed to all the experiments as well as data analysis. B.Y. performed in vivo mice studies. D.F. contributed to the analysis of crystal data and HRMS experiments. J.L. contributed to the in vitro drug resistance development study. H.S. contributed to the cell wall permeability study using a fluorescence microscopy. H.K. performed XRD experiments. S.D.H and M.K conceived and designed the research. R.S and M.K wrote the paper. All authors contributed to the editing of the manuscript and approved the final version.

## Competing interests
A provisional patent application related to this work was submitted (an application number 62891620 by R.S., B.Y., S.D.H. and M.K). The remaining authors declare no competing interests.
