## [Peer Review File · Communications Biology]

Reviewers' comments:

Reviewer #1 (Remarks to the Author):

The authors present a study on Naphthoquinone-derivative as a synthetic compound to overcome resistance in MRSA. The manuscript has written an objective and easily understandable to the reader. The theme is interesting and important study synthetic compounds that could be used for their therapeutic efficacy. The methodologies used remains well accepted. Suggestions for corrections were made. But, it was found some minor remarks in this paper:

1. In the part of " abstract"

} Line 3 : correct 1,4-naphthoquinone

2. Materiel and methods

} Your objective is to promote lawsone or derivatives of lawsone. You synthesise the lawsone or you purchased it? Clarify the origin.

} In the majority of your tests you consider practically just the compound 6c. I think it was wise to test all the compounds in order to have a clear idea on the mode of action of the Lawsone and its derivatives, knowing that 6c has shown the best activities.

} Can you justify the choice of murine strains

} In the part « In vivo antibacterial efficacy of 6c ...peritoneal infection ». You did not specify exactly the protocol that you adopted to do this test. We are missing details such as the number of lots used, how many 6c injections done, how many days of treatment ... Why didn't you test lawsone?

} In the part «For the induction of non-lethal peritoneal infection, a solution of .. two groups». How many mice and repetition the groups contain?.

} Determination of minimum inhibitory concentration (MIC). Which CLSI guidelines you used. There are several recommendations.

Synthesis and characterization of lawsone-derivatives.

Bacteria strains, growth media and antibiotics.

Mouse models of wound infection and peritoneal infection by MRSA.

Determination of minimum inhibitory concentration (MIC).

SEM imaging of MRSA and mouse red blood cells.

Cell membrane permeabilization assay.

Time-kill assay.

Iron rescue assay.

Measurement of intracellular irons in MRSA.

Measurements of intracellular ROS in MRSA.

Superoxide scavenging assay.

Mammalian cell viability assay.

Checkerboard assay

Biofilm inhibition assay

Biofilm staining with SYTO9 and PI

Hemolysis assay

In vitro assay for resistance development.

Lipophilicity and globularity determination.

the multifaceted modes of action of compound 6c, mediated by three distinctive mechanisms: (1) cell membrane damage, (2) chelation of intracellular iron ions, and (3) generation of intracellular ROS.

Reviewer #2 (Remarks to the Author):

Dear authors

The manuscript titled "Naphthoquinone-derivative as a synthetic compound to overcome resistance in MRSA" makes an important contribution to global public health, promoting the synthesis of promising compounds for the treatment of resistant infections that have plagued humanity for many years due to lack therapeutic strategies that are maintained over the years. I strongly suggest the acceptance of the manuscript, since it is very complete in the questions of initial pre-clinical investigation and brings prospects well punctuated for the development of the necessary steps. I only suggest that the authors review some points:

- In the methodology and insert important data for the replication of the methods, such as the identification of cell lines with their respective codes in the American Type Culture Collection, the indication of the documents chosen for the in vitro and in vivo tests, used according to the Clinical and Laboratory Standards Institute (CLSI) and the clear definition of which concentrations and vehicles were used in the experiments.

- In addition, in the tables I suggest that the authors indicate what "-" means, not accomplished? not determined?

Best Regards

Reviewer #3 (Remarks to the Author):

There is a need for novel approaches to the therapy of bacterial infections in general and staphylococcal infections in particular. The occurrence of strains of *Staphylococcus aureus* that are resistant to beta-lactam antibiotics (both in hospitals and in the community) and to the glycopeptide vancomycin (in hospitals) compromises conventional treatments. This paper describes a molecule which has novel inhibitory mechanisms viz membrane damage, chelation of Fe³⁺ intracellularly and generation of reactive oxygen species. The combined mechanisms of action cause a bactericidal effect at > 4 x the MIC. Furthermore high level resistance to the molecule did not develop during prolonged exposure as happens with a fluoroquinolone and vancomycin. The molecule lacks toxicity towards mammalian cells and results in reductions in CFUs after topical application to infected skin and systemic treatment of dissemination after IP injection

Comments on the microbiology

1. Many different MRSA strains cause infections. Different SCCmec elements have been acquired by horizontal gene transfer in several clonal lineages. This study employs a single historic clone of MRSA (the Iberian clone). The authors should at least report the 6c MIC for other lineages including the widespread USA300 community associated MRSA.

2. Biofilm experiments. The supplementary information showed that 6c inhibited the development of a biofilm in vitro. It is more relevant to determine if 6c has an effect on an established biofilm. Does it cause a reduction in biofilm density or the number of CFUs after a biofilm has become established ?

3. In Figure 3c the authors compare the ability of cells to take up propidium iodide as a measure of the loss of membrane integrity. The rationale for comparing 6c with vancomycin is not clear. A control with a molecule that is known to have membrane damaging activity against *S. aureus* should be used as well. It states that vancomycin has little effect on membrane integrity - a reference should be given. Interfering with cell wall peptidoglycan biosynthesis will indirectly cause membrane damage as the turgor pressure of the cytoplasm can no longer be contained

4. Leakage of intracellular contents following treatment with 6c is concluded from observation of electron micrographs. Leakage of a small molecule from the cytoplasm should be measured quantitatively.

5. Development of resistance during serial passage. Figure 5 shows data for the MRSA strain. According to MIC data in Table 1 this strain is resistant to a low level of orfloxacin (8 ug/ml). Please clarify. This does not show in Fig 5a

6. The authors cannot conclude that significant cross resistance does not develop. Figure 5b indicates that the MIC to 6c increases 4-fold in the vancomycin resistant mutant. This seems to be

a significant change although the Table does not indicate the number of replicates performed. This is particularly pertinent given that vancomycin "insensitivity" (VISA) is a common clinical phenomenon during prolonged treatment. Given that daptomycin is now used as an alternative to vancomycin and that daptomycin resistance also emerges during prolonged treatment it is pertinent to examine the MIC of 6c against VISA and daptomycin resistant strains

Response to reviewers

We thank the reviewers for their comments and suggestions and hope that our response will adequately address the perceived ambiguities in our manuscript. In response to the reviewers' comments we have made substantial changes to the manuscript, including suggested additional experiments. A point-by-point summary of these changes is listed below. The revised sentences are highlighted in yellow in the main text. We believe our responses to the reviewers' questions/comments have improved the manuscript significantly.

Reviewer #1

1. In the part of “ abstract” Line 3 : correct 1,4-naphthoguinone

Response: This is now corrected to “2-hydroxy-1,4-naphthoquinone”.

2. Your objective is to promote lawsone or derivatives of lawsone. You synthesise the lawsone or you purchased it? Clarify the origin.

Response: The underivatized lawsone molecule was purchased from Sigma-Aldrich and the derivatives of lawsone were synthesized using the protocol described in the methods. The origins of chemicals including underivatized lawsone are clarified in the section of “Synthesis and characterization of lawsone derivatives” in the Supplementary Methods.

3. In the majority of your tests you consider practically just the compound 6c. I think it was wise to test all the compounds in order to have a clear idea on the mode of action of the Lawsone and it derivatives, knowing that 6c has shown the best activities.

Response: Thank you for the suggestion. The rationale for conducting assays on the mode of action using only **6c** compound was that the **6c** only exhibited a potent antibacterial activity against various strains of *S. aureus* among all the lawsone derivatives tested. As shown in Table 1, the MIC values of other lawsone derivatives (6a, 6b, 6d, and 6c) did not exhibit suitable activity, even compared to underivatized lawsone and we thought that it would not be necessary to examine the mode of action for those other derivatives. We do not know yet why the **6c** could exhibit such a potent antibacterial activity while others cannot. As stated in the discussion, we reason that the degree of lipophilicity and molecular globularity of **6c** compound might confer an optimal state, which are neither too high nor too low, for achieving the enhancement of bacterial membrane permeability. The identification of membrane molecule in MRSA that might interact with **6c** is very important for our understanding on the mode of action of **6c** and this will be further examined in our future study.

4. Can you justify the choice of murine strains?

Response: We chose the C57BL/6J strain of mouse for our experimental models for local and systemic infections since the mice is the most widely used inbred strain as experimental model for in vivo study. The mouse is immune-competent and has been shown to represent

a suitable model for the study of *S. aureus* infections and the characterization of staphylococcal virulence factors (Kim et al. J Immunol Methods. 410: 88–99, 2014). Additionally, since our objective in this study was to test the antimicrobial efficacy of our **6c** compound using non-lethal MRSA infections, we reasoned that the immune-competent mice with normal innate immune activity would be appropriate for the purpose.

5. In the part « In vivo antibacterial efficacy of 6c ...peritoneal infection ». You did not specify exactly the protocol that you adopted to do this test. We are missing details such as the number of lots used, how many 6c injections done, how many days of treatment ... Why didn't you test lawsone?

Response: The suggested information of the protocol for in vivo mice study was stated in detail in the section of “Mouse models of wound infection and peritoneal infection by MRSA” in the Methods. The number of mice used per each experimental group is provided in the figure legend (Legend of Figure 6).

6. In the part «For the induction of non-lethal peritoneal infection, a solution of .. two groups». How many mice and repetition the groups contain?

Response: The information regarding the # of mice and repetition is provided in the Figure legend (in Figure 6). As stated, 8 mice were used for each group and the CFU counts from homogenized tissue collected from each mouse were run in duplicates.

7. Determination of minimum inhibitory concentration (MIC). Which CLSI guidelines you used. There are several recommendations.

Response: We conducted a MIC determination assay by following CLSI's M100 guidelines according to

“CLSI. Performance Standards for Antimicrobial Susceptibility Testing. *Clinical and Laboratory Standards Institute, Wayne, PA, USA. CLSI supplement M100, 27th ed.* (2017).”

This reference is cited in the section of “Determination of minimum inhibitory concentration (MIC)” in Methods.

Reviewer #2

1. In the methodology and insert important data for the replication of the methods, such as the identification of cell lines with their respective codes in the American Type Culture Collection, the indication of the documents chosen for the in vitro and in vivo tests, used according to the Clinical and Laboratory Standards Institute (CLSI) and the clear definition of which concentrations and vehicles were used in the experiments.

Response: The general information on the reproducibility of experiments, including the number of sample size and replicates are stated in the section of “Statistics and reproducibility” in Methods as follows.

“The studies for in vitro cell culture were run in triplicate (3 biological replicates and 3 technical replicates per each biological replicate). For in vivo mice studies, mice were randomly divided into two (for non-lethal systemic infection protocol) or three (for wound infection protocol) groups and CFU counts from the collected tissue of the mice were run in duplicates.”

The number of sample size for each assay is indicated in the figure legends. The identification *S. aureus* strains used for each assay is indicated in each section of the Methods and Results. The MIC determination assay was conducted by following the CLSI’s M100 guidelines (refer to our response to Reviewer 1- Q#7).

2. In addition, in the tables I suggest that the authors indicate what “-“ means, not accomplished? not determined?

Response: The “-“ means “not determined”. This is indicated in Table 1.

Reviewer #3

1. Many different MRSA strains cause infections. Different SCCmec elements have been acquired by horizontal gene transfer in several clonal lineages. This study employs a single historic clone of MRSA (the Iberian clone). The authors should at least report the 6c MIC for other lineages including the widespread USA300 community associated MRSA.

Response: Per the reviewer’s suggestion, we conducted an experiment by obtaining USA300 strain of MRSA from ATCC (ATCC BAA-1717, Strain designations: TCH1516[USA300-HOU-MR]) and determining the MIC of **6c** against this strain of MRSA. The MICs of conventional antibiotics (vancomycin, daptomycin, ofloxacin, and ciprofloxacin) and underivatized lawsone against the bacteria were also determined and compared with that of **6c** compound. The MIC of **6c** against the USA300 MRSA was measured to be the same as the MIC value of **6c** against other MRSA strain (ATCC BAA-44), with the MIC value of 1.25~2.5 µg/mL. This result is now updated in Table 1 as follows.

Table 1. MIC values, lipophilicity, and globularity for Lawsone derivatives against different strains of *Staphylococcus aureus*.

Compound	MIC ($\mu\text{g/mL}$)					ClogP ^a	ClogD _{7.4} ^b	Number of rotatable bond ^c	Globularity ^d	Molecular weight ^e (g/mol)
	SA (ATCC 29213)	SA (ATCC 6538)	VISA (ATCC 700699)	MRSA (USA300, ATCC BAA-1717)	MRSA (ATCC BAA-44)					
Lawsone	16	32	64	128	32	1.20	-1.7	0	0	174.16
6a	128	>128	128	>128	>128	3.12	0.6	2	0.096	264.28
6b	32	32	128	64	32	3.85	1.3	3	0.093	292.33
6c	1.25~1.9	0.6	1.25~2.5	1.25~2.5	1.25~2.5	4.47	1.9	5	0.12	320.39
6d	>128	>128	>128	>128	>128	6.23	3.7	7	0.086	348.44
6e	>128	>128	>128	>128	>128	7.45	4.9	9	0.060	376.50
Vancomycin	0.5	1	4	1	1	-2.04	-8.48	13	0.28	1449.27
Daptomycin	0.5	1	4	1	1	-4.07	-9.72	35	0.38	1620.69
Ofloxacin	0.25	0.25	16	0.25	8	0.83	-1.1	2	n.d	361.37
Ciprofloxacin	0.25	0.25	32	0.25	16	-0.29	-2.5	3	0.07	331.34

^aClogP = calculated LogP, generated using ACD/Percepta software. ^bClogD_{7.4} = calculated LogD at pH = 7.4, generated from ACD/Percepta software. ^cNumber of rotatable bond was generated from ACD/Percepta software. ^dGlobularity was calculated from Molecular Operating Environment (MOE). ^eMolecular weight was calculated from ChemDraw Professional 16.0. n.d: not determined

2. Biofilm experiments. The supplementary information showed that **6c** inhibited the development of a biofilm in vitro. It is more relevant to determine if **6c** has an effect on an established biofilm. Does it cause a reduction in biofilm density or the number of CFUs after a biofilm has become established?

Response: To address this, we conducted a biofilm destruction assay from an established biofilm formed by MRSA (ATCC BAA-44). The detailed method for this assay is now updated in the Method section (Biofilm destruction assay). In brief, MRSA biofilm was formed on 96-well plate under static culture condition for 24 h and then varying concentrations of **6c**, vancomycin, or daptomycin (4-512 $\mu\text{g/mL}$) were treated to the established biofilm. Then, the extent of biofilm destruction was assessed by quantifying biofilm mass using a crystal violet assay. Our results of biofilm inhibition assay (shown in Supplementary Figure 6a-c) showed the ability of **6c** in inhibiting the formation of MRSA biofilm at a dose of 5 μM (~1.5 $\mu\text{g/mL}$). However, our assay for the destruction of established biofilm shows that it requires much higher dose of **6c** (~256 $\mu\text{g/mL}$, Supplementary Figure 6d) to eradicate an established biofilm. This was also the case for vancomycin and daptomycin and the ability of **6c** to destruct an established biofilm was comparable to those of vancomycin and daptomycin. This result is updated in Supplementary Figure 6d (also shown below) and the related sentences in the Results section of main text are revised accordingly as below.

*“In addition to the potency of **6c** against planktonic phase of MRSA, **6c** could significantly suppress the formation of biofilm formed by MRSA at a dose of $1 \times \text{MIC}$ (5 μM or 1.5 $\mu\text{g/mL}$, Supplementary Figure 6a-c). However, it required much higher dose of **6c** (256 $\mu\text{g/mL}$) for the eradication of established biofilm, which was comparable to those of vancomycin and daptomycin (Supplementary Figure 6d).”*

Supplementary Figure 6d. Biofilm destruction assay. mean \pm s.d, n=7 independent experiments). *: $p < 0.05$, **: $p < 0.01$.

3. In Figure 3c the authors compare the ability of cells to take up propidium iodide as a measure of the loss of membrane integrity. The rationale for comparing 6c with vancomycin is not clear. A control with a molecule that is known to have membrane damaging activity against *S. aureus* should be used as well.

Response: Thank you for the suggestion. To further ascertain the ability of **6c** in directly causing membrane damage, we include daptomycin as a positive control in the comparing groups, which has been shown to exhibit an antimicrobial activity by causing membrane damage in bacteria including *S. aureus* (Hobbs et al., *J Antimicrob Chemother* **62**, 1003-8, 2008. and Muller et al. *PNAS* **113**, E7077-E7086, 2016). For this, MRSA (ATCC BAA-44) were treated with daptomycin under the same experimental conditions (dose and exposure time, 16 μ M for 2 hr) as **6c** and vancomycin and the morphological change of MRSA was imaged using a SEM. As shown in Fig.3c, the cell morphology of daptomycin-treated MRSA shows similar changes to those of **6c**, associated with the leakage of intracellular contents, but distinct from that of vancomycin-treated MRSA. We believe that this additional data supports the membrane damaging ability of 6c compound. In this revision, the related sentence for the rationale for using vancomycin and daptomycin is clarified in the main text as suggested.

“This result is comparable to that of daptomycin, a lipopeptide that targets bacterial cell membranes^{21,22}, used as a positive control.”

It states that vancomycin has little effect on membrane integrity - a reference should be given. Interfering with cell wall peptidoglycan biosynthesis will indirectly cause membrane damage as the turgor pressure of the cytoplasm can no longer be contained

Response: The rationale for including vancomycin in the experimental groups for Fig.3c was to validate whether 6c could directly cause the disruption of membrane integrity. We reasoned that, if this is the case, the membrane damage would occur in the early phase of 6c exposure to the MRSA. We agree with the reviewer that the ability of vancomycin to interfere with cell wall synthesis may indirectly cause membrane damage, but we reasoned that this would occur in the later phase of vancomycin treatment. This can be supported by published studies by Higgins et al. and Blackovich et al.

- Higgins, D.L. et al. Telavancin, a multifunctional lipoglycopeptide, disrupts both cell wall synthesis and cell membrane integrity in methicillin-resistant *Staphylococcus aureus*. *Antimicrob Agents Chemother* **49**, 1127-34, 2005.
- Blackovich, M.A.T. et al. Protein-inspired antibiotics active against vancomycin- and daptomycin-resistant bacteria. *Nat Commun* **9**, 22, 2018.

Both studies showed that vancomycin had little effect on cell membrane permeability against MRSA (ATCC 33591 and ATCC 43300) within the time frame tested (~60 min), which is consistent with our data that showed a negligible change in membrane permeability in MRSA (ATCC BAA-44) by vancomycin (Fig. 3b). The references are cited (ref #24 and #25 in this revision) to the associated sentence in the Results of main text as follows.

“with relatively little effect on membrane permeation^{24,25}.”

4. Leakage of intracellular contents following treatment with 6c is concluded from observation of electron micrographs. Leakage of a small molecule from the cytoplasm should be measured quantitatively.

Response: Thank you for the suggestion. We endeavored to quantify the leakage from the MRSA exposed to 6c, however we could not complete this assay due to the technical

difficulty in collecting leakage contents from the bacterial culture since the sample collection step could artificially induce the destruction of cell membrane. Although we could not demonstrate the presence of bacterial cellular leakage to **6c** treatment, we feel that our new data of SEM image from daptomycin treated-MRSA would support our conclusion.

5. Development of resistance during serial passage. Figure 5 shows data for the MRSA strain. According to MIC data in Table 1 this strain is resistant to a low level of ofloxacin (8 ug/ml). Please clarify. This does not show in Fig 5a

Response: The MIC of ofloxacin against MRSA (ATCC BAA-44) was measured to be 8 µg/mL. The legend of figure 5a is now revised to indicate this dose for clarification as follows.

*“The initial MIC values of 8 µg/mL, 1 µg/mL, and 2.5 µg/mL, were used for ofloxacin, vancomycin, and **6c** respectively.”*

6. The authors cannot conclude that significant cross resistance does not develop. Figure 5b indicates that the MIC to **6c** increases 4-fold in the vancomycin resistant mutant. This seems to be a significant change although the Table does not indicate the number of replicates performed. This is particularly pertinent given that vancomycin “insensitivity” (VISA) is a common clinical phenomenon during prolonged treatment. Given that daptomycin is now used as an alternative to vancomycin and that daptomycin resistance also emerges during prolonged treatment it is pertinent to examine the MIC of **6c** against VISA and daptomycin resistant strains.

Response: To address this, we conducted a MIC assay using VISA strain ATCC (ATCC 700699, MU50[NRS1] strain) purchased from ATCC. Our measurement also confirmed a vancomycin-intermediate characteristic of this strain, exhibiting a vancomycin MIC of 4 µg/mL against the bacteria. The strain exhibited resistance to other antibiotics including ofloxacin (MIC = 16 µg/mL) and ciprofloxacin (MIC = 32 µg/mL). Our new result for the MIC of **6c** against this VISA strain was measured to be the same as those against other MRSA strains (ATCC BAA-44 and ATCC-BAA 1717) at the value of MIC = 1.25~2.5 µg/mL, demonstrating the potency of **6c** against VISA as well as MRSA strains (Table 1, please refer to the table given above).

Regarding the reviewer’s suggestion to test the potency of **6c** against daptomycin-resistant strain of *S. aureus*, we used the same VISA strain of *S. aureus* (ATCC 700699) as an experimental model for this purpose, since the strain also exhibited a non-susceptibility to daptomycin (MIC = 4 µg/mL, refer to the Table 1 above). We could not obtain the daptomycin-resistant strain of MRSA from commercial sources and do not anticipate obtaining it from any other source at this time. Our **6c** compound exhibited a potency against this strain (with MIC value of 1.25-2.5 µg/mL, Table 1), we conclude that our **6c** compound is also effective against daptomycin non-susceptible strain of MRSA.

This result is updated in Table 1 and changes have been made in the main text as below

*“Among them, compound **6c** (hereafter **6c**) was found to exhibit the strongest antimicrobial activity against MRSA strains (ATCC BAA-44 and ATCC BAA-1717, MIC = 1.25 ~ 2.5 $\mu\text{g}/\text{mL}$), which was comparable to that of vancomycin and daptomycin (MIC = 1 $\mu\text{g}/\text{mL}$) (Table 1). The **6c** was also effective against vancomycin-intermediate *S. aureus* (VISA) (ATCC 700699) that is also non-susceptible to daptomycin (MIC = 4 $\mu\text{g}/\text{mL}$), with the same MIC value against MRSA.”*

REVIEWERS' COMMENTS:

Reviewer #3:

None